# Robust Learning via Golden Symmetric Loss of (un)Trusted Labels

## Abstract

Learning robust deep models against noisy labels becomes ever critical when today's data is commonly collected from open platforms and subject to adversarial corruption. The information on the label corruption process, i.e., corruption matrix, can greatly enhance the robustness of deep models but still fall behind in combating hard classes. In this paper, we propose to construct a golden symmetric loss (GSL) based on the estimated confusion matrix as to avoid overfitting to noisy labels and learn effectively from hard classes. GSL is the weighted sum of the corrected regular cross entropy and reverse cross entropy. By leveraging a small fraction of trusted clean data, we estimate the corruption matrix and use it to correct the loss as well as to determine the weights of GSL. We theoretically prove the robustness of the proposed loss function in the presence of dirty labels. We provide a heuristics to adaptively tune the loss weights of GSL according to the noise rate and diversity measured from the dataset. We evaluate our proposed golden symmetric loss on both vision and natural language deep models subject to different types of label noise patterns. Empirical results show that GSL can significantly outperform the existing robust training methods on different noise patterns, showing accuracy improvement up to 18% on CIFAR-100 and 1% on real world noisy dataset of Clothing1M.

## 1 Introduction

Diverse datasets collected from the public domain which power up deep learning models present new challenges – highly noisy labels. It is not only time consuming to collect labels but also difficult to ensure a consistent label quality due to various annotation errors (Patrini et al., 2017) and adversarial attacks (Goodfellow et al., 2015). The large capacity of deep learning models enables effective learning from complex datasets but also suffers from overfitting to the noise structure in the dataset. The curse of memorization effect (Jiang et al., 2018) can degrade the accuracy of deep learning models in the presence of highly noisy labels. For example, in (Zhang et al., 2017) the accuracy of AlexNet to classify CIFAR10 images drops from 77% to 10%, when there are randomly flipped labels.

Designing learning models that can robustly train on noisy labels is thus imperative. To distill the impact of noisy labels, the related work either filters out suspiciously noisy data, derives robust loss functions or tries to proactively correct labels. Symmetric Cross entropy Loss (SCL) is shown effective in combating label noise especially for hard classes by combing the regular with the reverse cross entropy. The former avoids overfitting and the latter is resilient to label noise. Given its promising results, there is yet to have a clear principle on how to weight the regular and reverse cross entropy terms, e.g., at different noise rates and patterns. In contrast, Distilling (Li et al., 2017) and Golden Loss Correction (GLC) (Hendrycks et al., 2018) advocate to use a small clean data to improve the estimated corruption matrix. Specifically, GLC trains the deep model on both a clean and noisy set, whose loss is corrected through the corruption matrix. While the clean set is evenly chosen from all classes, the corrupted labels may appear unevenly across classes depending on the noise pattern (Xiao et al., 2015). As the corrected loss of GLC does not differentiate the difficulty of classes, it may not learn those hard classes effectively.

We propose GSL constructing the golden symmetric loss that dynamically weights regular/reverse cross entropy and corrects the label prediction based on the estimated corruption matrix. Similar to

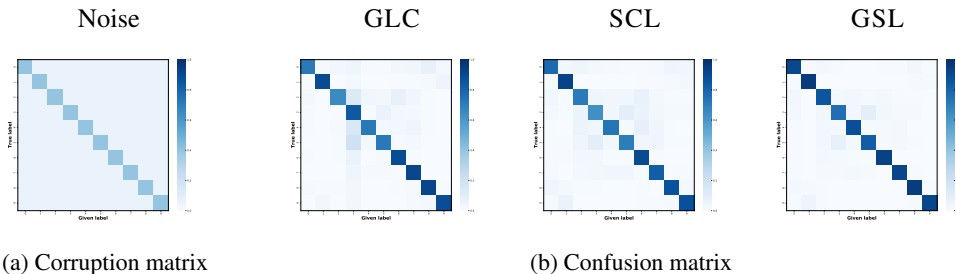

(a) Corruption matrix                    (b) Confusion matrix

Figure 1: Noise corruption matrix and confusion matrices of predictions for CIFAR-10 with 60% symmetric label noise.

GLC, GSL leverages clean data to estimate the corruption matrix which is used to correct labels and decide the weights of the golden symmetric loss. As such, GSL can effectively differentiate the difficulty level of classes by adjusting the weights and mitigate the impact of noise overfitting via the golden symmetric cross entropy. Specifically, we use the noise rate and noise diversity to adaptively tune the weights of modified cross entropy and reverse cross entropy. We prove that modified cross entropy by using confusion matrix is noise tolerant same as the reverse cross entropy.

**Motivation example.** We demonstrate the advantages and disadvantages of GLC and SCL, and the their combination (the proposed GSL) through the example of learning convolution networks on CIFAR-10 injected with 60% symmetric noise. The experimental setup is detailed in §6. Figure 1 shows the corruption matrix of the injected noise and the confusion matrices from the predictions of SCL, GLC, and GSL. Even if the injected noise is symmetric across all classes (see Figure 1a), prediction errors are distributed asymmetrically across the classes (see Figure 1b). Though GLC can achieve a lower average error rate than SCL (reflected in darker diagonal elements on average), it performs worse in hard classes, e.g., class 4 (cat) and class 6 (dog) (difference in blue shades across the diagonal elements). By setting up proper weights for two types of cross entropy, GSL is able to achieve both superior average and per class accuracy.

## 2 RELATED WORK

Enhancing the robustness of deep models against noisy labels is an active research area. The massive datasets needed to train deep models are commonly found corrupted, (Wang et al., 2018), severely degrading the achievable accuracy, (Zhang et al., 2017). The impact of label noise on deep neural networks is first characterized by the theoretical testing accuracy over a limited set of noise patterns (Chen et al., 2019). (Vahdat, 2017) suggest an undirected graph model for modeling label noise in deep neural networks and indicate symmetric noise to be more challenging than asymmetric. Solutions of the prior art can be categorized into three directions: (i) filtering out noisy labels: (Malach & Shalev-Shwartz, 2017; Han et al., 2018b; Yu et al., 2019; Wang et al., 2018); (ii) correcting noisy labels: (Patrini et al., 2017; Hendrycks et al., 2018; Li et al., 2017); and (iii) deriving noise resilient loss functions: (Ma et al., 2018; Konstantinov & Lampert, 2019).

**Noise Resilient Loss Function**. The loss function is modified to enhance the robustness to label noise by introducing new loss functions, (Ghosh et al., 2017; Wang et al., 2019), or adjusting the weights of noisy data instances, (Ren et al., 2018b; Konstantinov & Lampert, 2019; Ma et al., 2018). Mean Absolute Error (MAE) (Ghosh et al., 2017; Zhang & Sabuncu, 2018) and General Cross Entropy loss (Zhang & Sabuncu, 2018) are proposed as a noise resilient alternative but at the cost of slow convergence. To avoid overfitting to noise, D2L (Ma et al., 2018) uses the subspace dimensionality to assign weights to each data point, whereas Konstantinov (Konstantinov & Lampert, 2019) determines the loss weights based on the trustworthiness level of data sources. (Wang et al., 2019) propose symmetric cross-entropy loss that combines a new term of reverse cross entropy with traditional cross entropy via constant weights on both terms. Meta-Weight-Net (Shu et al., 2019) re-weights samples during optimizing loss function in the training process by using a multi-layer perceptron to predict the weight of each sample. With the same perspective, (Ren et al., 2018a) uses the similarity of samples to the clean instances in the validation set for re-weighting them in loss function.

**Label correction**. To avoid the data reduction caused by filtering, label correction methods adjust the predicted/given labels by using only noisy labels (Patrini et al., 2017; Tanaka et al., 2018) or jointly with a small fraction of trusted data (Veit et al., 2017; Han et al., 2018a; Li et al., 2017; Hendrycks et al., 2018). Reed et al. train the classifier by the "new" labels combining the raw and predicted labels without access to label ground truth. (Patrini et al., 2017) estimate the noise confusion matrix by first training a classifier on the noisy labels and then using the softmax probabilities. (Veit et al., 2017) acquire human-verified labels to train a cleaning network for correcting noisy labels of multi-label classification problems. (Han et al., 2018a) estimate the noise transition probability by incorporating human assistance. (Li et al., 2017) and (Hendrycks et al., 2018) leverage a small set of clean data to estimate noise corruption matrix from the clean and noisy sets, respectively. DivideMix (Li et al., 2020) is a semi-supervised method, including two networks and Gaussian Mixture Model for sample selection.

The proposed GSL combines resilient loss function and label correction by curating a small fraction of trusted data. We solicit a subset of informative data instances to estimate the confusion matrix and provide a minimum supervision on noisy labels. We also provide a heuristic to adaptively tune the weights of golden symmetric loss according to the noise characteristics of the dataset.

## 3 GOLDEN SYMMETRIC LOSS

Consider the classification problem having dataset $\tilde{\mathcal{D}} = \{(\boldsymbol{x}_n, \tilde{y}_n)\}_{n=1}^N$ where $\boldsymbol{x}_n \in \mathcal{X} \subset \mathbb{R}^d$ denotes the $n^{th}$ observed sample, and $\tilde{y}_n \in \mathcal{Y} := \{1, ..., K\}$ the corresponding given label over $K$ classes. Hereon $n$ is ignore for the simplicity. $\tilde{y}$ is affected by label noise. The label corruption process is characterised by a corruption matrix $C_{ij} = P(\tilde{y} = j | y = i)$ for $i = 1, \ldots, K$ and $j = 1, \ldots, K$ where $y$ is the true label. Synthetic noise patterns are expressed as a label corruption probability $\varepsilon$ plus a noise label distribution. Let $g(\cdot, \boldsymbol{\theta})$ denote a neural network-based classifier parameterized by $\boldsymbol{\theta}$. For each data point $\boldsymbol{x}$, $f(\cdot, \boldsymbol{\theta})$ predicts the probability for each class label $k$: $p(k|\boldsymbol{x}) = \frac{e^{z_k}}{\sum_{j=1}^K e^{z_j}}$ where $z_j$ are the logits.

**Symmetric Cross Entropy**. Let $q(k|\boldsymbol{x})$ denote the ground truth probability distribution over the $K$ class labels where $q(k|\boldsymbol{x}) = 1$ for $k$ equal to the true class $y$ and $q(k|\boldsymbol{x}) = 0$ for all $k \neq y$. The cross entropy loss ($\ell_{ce}$) and reverse cross entropy loss[1] ($\ell_{rce}$) for sample $\boldsymbol{x}$ are:

$$\ell_{ce} = -\sum_{k=1}^K q(k|\boldsymbol{x}) \log p(k|\boldsymbol{x}), \quad \ell_{rce} = -\sum_{k=1}^K p(k|\boldsymbol{x}) \log q(k|\boldsymbol{x}). \tag{1}$$

(Wang et al., 2019) combine cross entropy and reverse cross entropy into the symmetric cross entropy:

$$l_{sl} = \alpha \, \ell_{ce} + \beta \, \ell_{rce}. \tag{2}$$

where $\alpha$ and $\beta$ are hyperparameters. On the one hand cross entropy loss is not robust to noise (Ghosh et al., 2017) but achieves good convergence (Zhang & Sabuncu, 2018). On the other hand reverse cross entropy is tolerant to noise (Wang et al., 2019).

**Estimating Noise Corruption Matrix**. We estimate the noise corruption matrix as in (Hendrycks et al., 2018). The method fosters training a first classifier $g(\cdot, \Theta)$ on noisy data to approximate the elements $C_{ij}$ of the noise corruption matrix via a small fraction of trusted data $\mathcal{D}$ with known true label $y$. Practically given $A_i$ the subset of trusted data with label of class $i$ $\{A_i \subset \mathcal{D} : y = i\}$, the elements of $\boldsymbol{C}$ can be approximated by:

$$\hat{\boldsymbol{C}}_{ij} = P(\tilde{y} = j | y = i) \approx \frac{1}{|A_i|} \sum_{x \in A_i} g(\tilde{y} = j | \boldsymbol{x}, \Theta) \tag{3}$$

where $g(\tilde{y} = j | \boldsymbol{x}, \Theta)$ denotes predicted probability of $\boldsymbol{x}$ having class label $j$. That is $\hat{\boldsymbol{C}}_{ij}$ is computed as the mean predicted probability of class $j$ for all trusted data points having true label of class $i$.

**Training with Corrected Labels**. Let $\hat{C}$ be the estimated noise corruption matrix. Using the method in (Patrini et al., 2017), we increase the noise resilience by correcting the predictions of the

---

[1]To avoid problems with the logarithm, zero values of $q$ are replaced by a small positive value, i.e. $10^{-4}$.

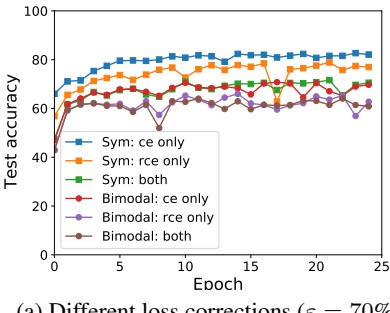 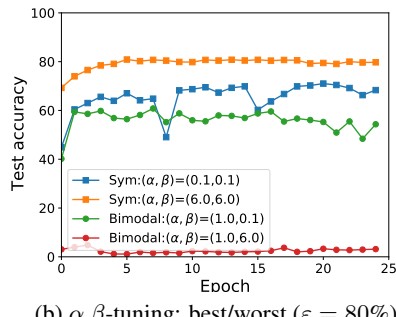

(a) Different loss corrections ($\varepsilon = 70\%$)  (b) $\alpha,\beta$-tuning: best/worst ($\varepsilon = 80\%$)

Figure 2: Impact of loss correction and $\alpha,\beta$-tuning on a 2-layer FC network trained on Twitter data.

classifier using $\hat{C}$. Let $\hat{p}$ be the corrected predicted probabilities $\hat{p} = \hat{C}^T p$, i.e. for data point $\boldsymbol{x}$: $\hat{p}(k|\boldsymbol{x}) = \sum_{i=1}^{K} \hat{C}_{ik} p(i|\boldsymbol{x})$ for $k = 1, \ldots, K$. We enhance the regular cross entropy term. Applying the prediction correction to both terms holds lower benefits. We evaluate this empirically with extensive experiments on datasets of text, i.e. Twitter in Figure. 2a, and images, i.e. CIFAR10 and CIFAR100 in Appendix A. Experiment details can be found in §6. We consider different datasets, noise rates, noise types and fractions of trusted data. We see that in all cases, except one with a difference $< 0.3\%$, correcting only the cross entropy (*ce-only*) holds better results than correcting only the reverse cross entropy (*rce-only*) or correcting *both*. Focusing on Figure. 2a, *ce-only* improves accuracy by up to 5% and 8% for bimodal and symmetric noise, respectively. In case of CIFAR-10 and CIFAR-100 datasets the improvements are more pronounced with up to 11% and 50% respectively.

**Golden Symmetric Loss**. Towards a more effective and robust learning we propose to leverage the estimated noise corruption matrix $\hat{C}$ to tune the two loss terms based on the observed noise pattern. $\alpha$ and $\beta$ can significantly impact the final model accuracy. Tuning these parameters is no mean feat as different datasets affected by different noise patterns benefit from different optimal values (Wang et al., 2019). Again we show this behavior by training a 2-layer FC neural network on the Twitter dataset under eleven different $(\alpha, \beta)$ combinations and two noise patterns with 80% noise. Figure. 2b reports for each noise pattern the evolution over the training epochs of the test accuracy for the best and worst $(\alpha, \beta)$-pair. For bimodal noise even with a small number of trials, the impact of $(\alpha, \beta)$ ranges from an accuracy close to 60% all the way down to almost 0%. Moreover only few (two out of eleven) $(\alpha, \beta)$-pairs hold accuracy close to 60%. For symmetric noise the tuning impact is lower (limited between 70% and 80%) but the best and worst $(\alpha, \beta)$-pair differ from the bimodal noise case. This underlines both the importance and difficulty of tuning $(\alpha, \beta)$. Motivated by the high impact of $\alpha$ and $\beta$, we propose to dynamically weight the regular and reverse cross entropy terms. Let $A()$ and $B()$ be weighting functions mapping $\hat{C} \to \mathbb{R}$ we define a new loss function:

$$\ell_{GSL} = A(\hat{C})\, \ell_{ce} + B(\hat{C})\, \ell_{rce} \tag{4}$$

We call this new loss function golden symmetric loss. $A()$ and $B()$ should capture not only the intensity of the noise pattern, but also the diversity of the noise pattern (see Figure. 2b).

**Determining Weights of Golden Symmetric Loss** ($A()$ **and** $B()$). In general the more intense and asymmetric the noise pattern, the lower the weight values should be. Since the final loss function learns from both dirty and clean data (see next paragraph), lower values of $\alpha$ and $\beta$ reduce the influence of dirty data over the one of clean data. Hence, we design $A()$ and $B()$ to capture both noise intensity and diversity. The intensity is given by the noise rate $\varepsilon \in [0, \ldots, 1]$, i.e. one minus the average of the diagonal elements of $\hat{C}$. The diversity is measured via Jain's fairness index $J(x_1, x_2, \ldots, x_n) \triangleq (\sum_{i=n}^{n} x_i)^2 / n \sum_{i=n}^{n} x_i^2$. We choose $J$ because it bounds the diversity on a similar scale as $\varepsilon$ between 1 (all equal, full symmetry) down to $1/n$ (highest asymmetry). We apply $J$ on all the $K(K-1)$ noise, i.e. off the diagonal, elements of $\hat{C}$:

$$J = \frac{(\sum_{i=1}^{K} \sum_{j=1, j \neq i}^{K} \hat{C}_{ij})^2}{K(K-1) \sum_{i=1}^{K} \sum_{j=1, j \neq i}^{K} \hat{C}_{ij}^2} \tag{5}$$

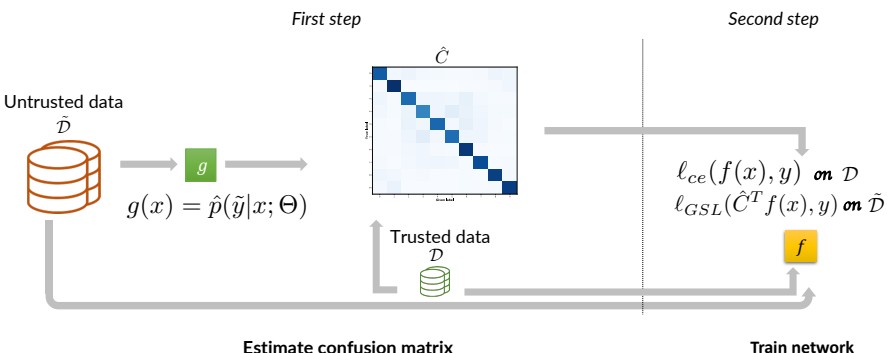

Figure 3: Training process of GSL divided into two steps.

For symmetric noise $J = 1$, the more asymmetric the smaller $J$. Final weights proportional to $J, \varepsilon$.

**Putting It All Together**. As a final step, to maximize the utility of the trusted data, we foster $\mathcal{D}$ as additional trusted training data for $f()$. Since $\mathcal{D}$ contains the true labels $y$ no prediction correction is applied. Hence, the overall loss function for data points from both $\mathcal{D}$ and $\tilde{\mathcal{D}}$ is:

$$
l = \begin{cases} -A(\hat{\boldsymbol{C}}) \sum_{k=1}^{K} q(k|\boldsymbol{x}) \log(\sum_{i=1}^{K} \hat{\boldsymbol{C}}_{ik} p(i|\boldsymbol{x})) - B(\hat{\boldsymbol{C}}) \sum_{k=1}^{K} p(k|\boldsymbol{x}) \log q(k|\boldsymbol{x}), & x \in \tilde{\mathcal{D}} \\ -\sum_{k=1}^{K} q(k|\boldsymbol{x}) \log p(k|\boldsymbol{x}), & x \in \mathcal{D}. \end{cases}
\tag{6}
$$

Figure. 3 summarises visually the training process divided into two main steps: (i) estimating noise corruption matrix through the first network $g$ trained on untrusted dataset $\tilde{\mathcal{D}}$ and (ii) training classifier $f$ on both untrusted $\tilde{\mathcal{D}}$ and trusted $\mathcal{D}$ through the golden symmetric loss.

## 4 THEORETICAL ANALYSIS

We prove that the cross entropy loss with label correction is noise tolerant under the definition put forth by (Ghosh et al., 2017; Manwani & Sastry, 2013) and extending prior art results. Let the risk of classifier $f$ and loss function $\ell_{ce}$ under clean labels be $R(f) = \mathbb{E}_{\boldsymbol{x},y} \ell_{ce}(f(\boldsymbol{x}), y)$ and the *risk* under noise rate $\varepsilon$ be $R_\varepsilon(f) = \mathbb{E}_{\boldsymbol{x},\tilde{y}} \ell_{ce}(f(\boldsymbol{x}), \tilde{y})$. $\mathbb{E}$ indicates the expectation taken over the random variables indicated as its subscripts. With prediction correction via $\boldsymbol{C}$, the risk becomes $R_\varepsilon(f, \boldsymbol{C}) = \mathbb{E}_{\boldsymbol{x},\tilde{y}} \ell_{ce}(\boldsymbol{C}^T f(\boldsymbol{x}), \tilde{y})$. Let $f^*$ and $f_\varepsilon^*$ be the global minimizers of $R(f)$ and $R_\varepsilon(f)$, respectively, and $\boldsymbol{C^*} = p(\tilde{y}|y)$ and $\hat{C}$ be the true and estimated noise confusion matrices, respectively.

**Theorem 4.1.** *In a multi-class classification problem, $\ell_{ce}$ with prediction correction is noise tolerant under symmetric label noise if the noise rate $\varepsilon < \frac{K-1}{K-\frac{\Delta\mathcal{A}}{\Delta R}}$, where $\Delta\mathcal{A} = \sum_{k=1}^{K} \ell_{ce}(\boldsymbol{C^*}^T f(\boldsymbol{x}), k) - \sum_{k=1}^{K} \ell_{ce}(\hat{\boldsymbol{C}}^T f(\boldsymbol{x}), k)$, and $\Delta R$ is the difference of risk minimization between optimal classifier and $f$. And $\ell_{ce}$ with prediction correction is also noise tolerant under flip noise when noise rate $\varepsilon_{yk} \leq (1 + \frac{\Delta\mathcal{W}_y}{\Delta\mathcal{W}_k}) - \varepsilon_y(1 + \frac{\Delta\mathcal{W}_y}{\Delta\mathcal{W}_k})$ where $\varepsilon_k$ and $\varepsilon_{yk}$ are the correct and flipped class probabilities, respectively.*

The proof is based on the risk minimization framework aiming to show under which condition $R_\varepsilon(f^*, \boldsymbol{C^*}) - R_\varepsilon(f, \hat{\boldsymbol{C}}) \leq 0$, i.e. the loss function is robust to noise. The detailed steps of the proof can be found in Appendix C. The condition $\varepsilon < \frac{K-1}{K-\frac{\Delta\mathcal{A}}{\Delta R}}$ is a generalization of the previous bound $\varepsilon < \frac{K-1}{K}$ by (Ghosh et al., 2017). Without label correction $\Delta\mathcal{A} = 0$ which corresponds to the previous result. Label correction improves the robustness by allowing higher noise rates, i.e. with label correction $\frac{\Delta\mathcal{A}}{\Delta R} \geq 0$ and hence $\frac{K-1}{K} \leq \frac{K-1}{K-\frac{\Delta\mathcal{A}}{\Delta R}}$. Similar observations hold for flip noise bound.

## 5 EXPERIMENTAL SETUP

**Dataset, Architecture and Parameters**. We consider two types of datasets: vision and text analysis. For vision, we use convolution neural networks (CNN) to classify CIFAR-10 and CIFAR-100 with injected label noise and Clothing1M as real world noisy dataset. For text, we use fully connected neural networks to classify noisy Twitter and Stanford Sentiment Treebank (SST). In principle, we use the same network architecture on all comparative approaches across different noise resilience techniques. In addition, we test the original network from the respective papers too and report the best results among the two.

- **CIFAR-10** (Krizhevsky et al., 2009). It contains 60K images classified into 10 classes: 50K as a training set and 10K as validation set. We use the architecture of Wide-ResNet by (Zagoruyko & Komodakis, 2016) with depth 28 and a widening factor 10 and train it with SGD with Nesterov momentum and a cosine learning rate schedule (Loshchilov & Hutter). For GSL, we first train $f$ for 75 epochs to obtain the noise corruption matrix. Then we train $g$ for 120 epochs.

- **CIFAR-100** (Krizhevsky et al., 2009). It contains 60K images classified into 100 classes: 50K as training set and 10K as the validation set. We use the same Wide-ResNet architecture used for CIFAR-10. For GSL, we train the $f$ and $g$ networks for 75 and 200 epochs, respectively.

- **Clothing1M** (Xiao et al., 2015). This is a real world dataset with label noise. It includes images scrapped from the Internet classified into 14 categories. We resize and crop each image to $224 \times 224$ pixels. This dataset contains 47K and 10K images for training and testing, respectively. These two sets have both given (scrapped) and true (human-checked) labels. We use ResNet-50 pretrained with ImageNet and further train for 10 epochs with batch size 32, SGD optimizer, momentum 0.9, weight decay $10^{-3}$, and learning rate $10^{-3}$ which is divided by 10 after 5 epochs.

- **Twitter** (Gimpel et al., 2011). The Twitter dataset includes 1,827 tweets annotated with 25 POS tags split in 1000 tweets as training set, 327 tweets as development set and 500 tweets as test set. We add development set to training set, and consider it as a training set. We use a 2-layer fully connected network with 256 hidden neurons each and GELU nonlinearity as activation function. We train $f$ with Adam for 15 epochs with batch size 64 and learning rate of 0.001. We train $g$ for 25 epochs. To regularize all linear output layer, we use $\ell_2$ weight decay with $\lambda = 5 \times 10^{-5}$.

- **Stanford Sentiment Treebank** (Socher et al., 2013). The SST dataset includes single sentence movie reviews. We use the 2-class version, including 6911 reviews in the training set, a development set with 872 reviews, and 1821 reviews in the test set. We augment the training set by using development set. We learn 100-dimensional word vectors from scratch for a vocab size of 10000. We train a word-averaging model with an affine output layer using Adam optimizer for 5 epochs for network $f$ and 10 epochs for network $g$. The batch size and learning rate are 50 and 0.001, respectively. To regularize all linear output layer, we use $\ell_2$ weight decay with $\lambda = 1 \times 10^{-4}$.

**Noise Corruption**. We consider symmetric noise and two different asymmetric noises, namely flip and bimodal. Symmetric noise corrupts the true label into a random other labels with equal probability based on the noise rate. The flip noise is generated by flipping the original label to a paired other class with a specific probability. The bimodal noise imitates targeted adversarial attacks (Goodfellow et al., 2015). Specifically, the true labels are corrupted into two neighborhoods centered on two targeted classes, each of which follows truncated normal distribution, $\mathcal{N}^T(\mu, \sigma, a, b)$. $\mu$ specifies the target and $\sigma$ controls the spread. $a$ and $b$ simply define the class label boundaries. For CIFAR-10 we target class 3 and 7, for CIFAR-100 class 30 and 70, for Twitter class 6 and 18, and for SST class 0 and 1. Instead, Clothing1M is already affected by real world label noise and left untouched.

## 6 EVALUATION

In this section, we empirically compare GSL against state of the art noise resilient networks on noisy vision and text data. We aim to show the effectiveness of GSL via testing accuracy on diverse and challenging noise patterns. Our target evaluation metric is the accuracy achieved on the clean testing set, i.e. not affected by noise.

Table 1: Vision analysis: test accuracy(%) of real-world noisy Clothing1M, and CIFAR10/CIFAR100 corrupted with 30% and 60% noise for different noise resilient networks. Best results in bold.

| CIFAR-10 | | | | | | | |
|---|---|---|---|---|---|---|---|
| Noise Rate | Noise Pattern | GSL | GLC | SCL | FORWARD | BOOTSTRAP | SGFORWARD | CO-TEACHING+ |
| 30% | Sym. | **92.75** | 89.60 | 83.56 | 74.06 | 75.78 | 91.03 | 76.53 |
| 30% | Bimodal | **92.89** | 89.20 | 82.83 | 73.84 | 75.74 | 91.24 | 74.97 |
| 30% | Flip | 90.66 | 91.09 | 81.52 | 78.55 | 78.90 | **90.81** | 79.75 |
| 60% | Sym. | **88.94** | 81.60 | 73.67 | 54.37 | 59.15 | 88.10 | 63.84 |
| 60% | Bimodal | **87.85** | 84.80 | 59.84 | 46.70 | 47.10 | 86.62 | 58.09 |
| 60% | Flip | **84.97** | 80.33 | 56.65 | 59.45 | 59.29 | 82.44 | 65.08 |

| CIFAR-100 | | | | | | | |
|---|---|---|---|---|---|---|---|
| Noise Rate | Noise Pattern | GSL | GLC | SCL | FORWARD | BOOTSTRAP | SGFORWARD | CO-TEACHING+ |
| 30% | Sym. | **75.80** | 61.80 | 58.66 | 40.80 | 43.30 | 72.63 | 53.94 |
| 30% | Bimodal | **76.38** | 61.40 | 47.62 | 45.22 | 42.46 | 73.87 | 55.22 |
| 30% | Flip | 75.57 | 75.23 | 55.23 | 54.70 | 55.08 | **76.03** | 58.86 |
| 60% | Sym. | **68.31** | 51.30 | 29.63 | 19.83 | 16.90 | 67.01 | 38.07 |
| 60% | Bimodal | **65.74** | 48.90 | 30.01 | 19.57 | 10.47 | 63.39 | 34.09 |
| 60% | Flip | **69.21** | 66.96 | 40.61 | 38.63 | 38.14 | 67.93 | 40.99 |

| Clothing1M | | | | | | | |
|---|---|---|---|---|---|---|---|
| Noise | GSL | GLC | SCL | FORWARD | BOOTSTRAP | SGFORWARD | CO-TEACHING+ |
| Real World | **74.86** | 73.91 | 70.78 | 70.04 | 67.87 | 73.96 | 70.33 |

## 6.1 VISION ANALYSIS

We compare GSL against four noise resilient networks from the state of the art: GLC (Hendrycks et al., 2018), SCL (Wang et al., 2019), FORWARD (Patrini et al., 2017), BOOTSTRAP (Reed et al.) and CO-TEACHING+ (Yu et al., 2019). In addition, we extend FORWARD by adding reverse cross-entropy based on SCL and loss correction through the confusion matrix same as GLC, called SGFORWARD. For training GSL, CO-TEACHING+, SGFORWARD and GLC, we use PyTorch v1.4.0. For all other methods, we use Keras v2.2.4 and Tensorflow v1.13.0.We assume 10% of trusted data is available for GSL, GLC and SGFORWARD. Table 1 summarizes the testing accuracy for all combinations of noise patterns and comparative approaches.

For CIFAR-10, GSL achieves the highest accuracy among all resilient networks except for flip noise with 30% noise rate. SGFORWARD is the closest rival to GSL because both use the same mechanism in the loss function. Besides, GSL has 2 to 8% higher accuracy than GLC, demonstrating the benefit of introducing symmetric cross-entropy, especially in high noise rates. In terms of comparison between GSL and SCL, the accuracy difference is even more visible, implying the benefit of using corruption matrix to assign weights on two terms in symmetric cross-entropy. We note that SCL uses an 8-layer CNN with 6 convolutional layers followed by 2 fully connected layers instead of a Wide ResNet because of the superior results. SCL performs particularly worse in 60% bimodal noise because this is a more challenging pattern and has no access to the corruption matrix. Moreover, our method can still obtain 11 to 30% higher test accuracy than CO-TEACHING+ that uses two deep networks concurrently.

CIFAR-100 is more challenging than CIFAR-10 due to the larger number of classes. GSL achieves the highest accuracy except for flip noise with 30% rate, and SGFORWARD is the second best result among other competitors. Although for flip noise with 30% rate SGFORWARD performs better than GSL, the improvement of GSL is more significant than SGFORWARD compared to the CIFAR-10 dataset. The largest difference (more than 2%) in accuracy between the GSL and SGFORWARD methods is with bimodal noise. In case of 60% symmetric noise, GSL achieves the accuracy of 68%, whereas GLC and SCL trail far behind. Moreover, given the difficulty of training a robust classifier for CIFAR-100 with 60% label noise, it is worth mentioning that SCL can achieve similar performance as GLC that is given 10% of trusted data in case of 30% symmetric noise. This also indicates the effectiveness of symmetric cross entropy in learning hard classes even without trusted

data. However, when facing extremely noisy labels and patterns, the small amount of trusted data can greatly improve the robustness of the classifier but not necessarily the symmetric cross entropy.

Seen from the high accuracy compared to GLC and SCL, GSL effectively uses the trusted data to correct symmetric cross entropy loss and improve the learning on the hard classes. GSL performs slightly better with symmetric noise than with bimodal and flip noise that is more challenging for CIFAR-10. In the CIFAR-100, GSL works better on the asymmetric noise rather than symmetric.

For Clothing1M dataset, as shown in Table 1, GSL obtains the highest test accuracy compared to other methods. Same as CIFAR-10 and CIFAR-100, SGFORWARD achieves a relatively good performance. The difference between GSL and SCL comes from the effectiveness of corruption matrix that makes the regular cross entropy robust.

Table 2: Text analysis: average accuracy (%) of variants combining loss correction and symmetric cross entropy. Results averaged across entire range of noise rates $[0, 100]$. Best accuracy in bold.

| | Noise Pattern | Percent Trusted | GSL | GLC | GFORWARD | TMATRIX | SGFORWARD | STMATRIX |
|---|---|---|---|---|---|---|---|---|
| **Twitter** | Sym. | 1 | **79.27** | 66.46 | 53.36 | 76.69 | 78.76 | 78.95 |
| | Sym. | 5 | **81.85** | 77.18 | 59.47 | 79.86 | 81.24 | 81.30 |
| | Bimodal | 1 | 75.74 | 67.05 | 52.72 | **77.92** | 75.64 | 76.64 |
| | Bimodal | 5 | **84.14** | 78.39 | 60.57 | 80.87 | 80.30 | 80.39 |
| | Flip | 1 | 75.52 | 83.21 | 39.37 | **86.21** | 73.83 | 73.30 |
| | Flip | 5 | 80.35 | 85.81 | 48.64 | **86.34** | 79.79 | 80.07 |
| **SST** | Sym. | 0.1 | **74.38** | 73.21 | 72.20 | 73.73 | 72.20 | 73.84 |
| | Sym. | 1 | **75.89** | 72.88 | 73.40 | 75.39 | 72.89 | 75.12 |
| | Bimodal | 0.1 | **74.81** | 74.80 | 72.77 | 74.07 | 72.73 | 74.06 |
| | Bimodal | 1 | **74.66** | 74.61 | 72.07 | 74.24 | 71.76 | 73.90 |
| | Flip | 0.1 | **75.31** | 74.13 | 49.48 | 74.88 | 49.47 | 74.95 |
| | Flip | 1 | **76.44** | 74.59 | 50.29 | 76.27 | 49.81 | 75.76 |

### 6.1.1 TEXT ANALYSIS

We evaluate GSL on text datasets of Twitter and SST, against resilient networks that leverage corruption matrix, namely GLC and FORWARD. Both GSL and GLC use the trusted data for estimating the corruption matrix, wheres the original FORWARD (Patrini et al., 2017) relies solely on the noisy data. As the proposed loss of golden symmetric cross entropy is general and can be combined with different resilient networks, we hence use following four variations of loss correction and symmetric cross entropy on the existing work:

- Forward gold (GFORWARD): we replace the estimation of the corruption matrix by the identity matrix on trusted samples and apply loss correction through the confusion matrix.
- True corruption matrix (TMATRIX): we directly use the true corruption matrix and apply loss correction through it.
- Forward gold with symmetric cross entropy (SGFORWARD): we extend the corrected loss of GFORWARD to the corrected symmetric cross entropy as in the GSL.
- True corruption with symmetric cross entropy (STMATRIX): we apply golden symmetric cross entropy and the true corruption matrix instead of the estimated matrix.

We extensively evaluate GSL, GLC GFORWARD, TMATRIX, SGFORWARD, and STMATRIX on Twitter and SST, with label corruption ranging from 0% to 100%. We also vary the percentage of trusted data among 1% and 5%. We summarize the average accuracy across 11 noise rates in Table 2.

**Twitter**. GSL consistently achieves the highest average accuracy in most cases. Compared to GLC, GSL has significant higher accuracy for Twitter corrupted with symmetric and bimodal noises, but the difference diminishes with increasing amounts of trusted data. When the percent of trusted data is low, say, 1%, GLC is unable to estimate the corruption matrix accurately nor to correct the loss, seen by the difference between GLC and TMATRIX.

**SST**. Here, the classification involves only two classes and turns out to be less challenging than the Twitter case. The difference among the different comparative approaches considered is smaller than

for Twitter. For instance, though GSL consistently achieves the best average accuracy in almost all cases, the difference between GSL and GLC is around 1-3%. Again, we see that GSL visibly outperforms GLC on low amounts of trusted data because of using cross entropy and the difference among them becomes limited. We note that TMATRIX and GFORWARD collapse under Flip noise.

## 7 CONCLUSION

To enhance the robustness of deep models against by label noise, we propose GSL that features on correcting the symmetric cross entropy loss by the noise corruption matrix. GSL uses a small fraction of trusted data to accurately estimate the corruption matrix, and further determine the weights applied on regular and reverse cross entropy. GSL learns deep networks from trusted samples through regular cross entropy and from untrusted noisy samples through golden symmetric cross entropy. We prove that the cross entropy corrected by the corruption matrix is noise robust. To adapt to noise patterns of dataset, we heuristically set the weights of golden symmetric loss based on the corruption matrix. We extensively evaluate GSL on vision and text analysis under diversified noise rates and patterns. Evaluation results show that GSL can achieve a remarkable accuracy improvement, i.e., from 2 to 18% on CIFAR benchmarks and real world noisy data, compared to methods that either correct loss or leverage symmetric cross entropy.

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

Table 3: Accuracy (%) of different gold fraction on CIFAR-10

| Noise rate = 30% | | | | | | | | | |
|---|---|---|---|---|---|---|---|---|---|
| Label correction | Bimodal | | | Symmetric | | | Flip | | |
| | 5% | 10% | 15% | 5% | 10% | 15% | 5% | 10% | 15% |
| *ce only* | **90.06** | **91.50** | **92.53** | **90.17** | **91.70** | **92.50** | **89.96** | **91.27** | **92.58** |
| *rce only* | 84.67 | 88.46 | 89.30 | 84.82 | 88.11 | 89.98 | 81.70 | 86.79 | 88.40 |
| *both* | 83.25 | 87.45 | 89.30 | 83.43 | 87.70 | 89.46 | 78.00 | 85.42 | 88.21 |
| Noise rate = 60% | | | | | | | | | |
| Label correction | Bimodal | | | Symmetric | | | Flip | | |
| | 5% | 10% | 15% | 5% | 10% | 15% | 5% | 10% | 15% |
| *ce only* | **83.06** | **88.43** | **90.52** | **85.79** | **89.19** | **90.19** | **80.03** | **82.64** | **85.80** |
| *rce only* | 81.73 | 86.86 | 89.07 | 81.54 | 86.63 | 88.98 | 68.29 | 80.22 | 84.35 |
| *both* | 79.83 | 85.79 | 88.63 | 80.27 | 86.22 | 88.53 | 63.63 | 82.19 | 83.18 |

Table 4: Accuracy (%) of different gold fraction on CIFAR-100

| Noise rate = 30% | | | | | | | | | |
|---|---|---|---|---|---|---|---|---|---|
| Label correction | Bimodal | | | Symmetric | | | Flip | | |
| | 5% | 10% | 15% | 5% | 10% | 15% | 5% | 10% | 15% |
| *ce only* | **71.66** | **73.90** | **75.20** | **71.69** | **74.24** | **75.11** | **74.78** | **75.46** | **77.01** |
| *rce only* | 25.90 | 61.44 | 67.15 | 26.15 | 61.19 | 67.14 | 23.75 | 59.68 | 65.33 |
| *both* | 23.37 | 57.52 | 64.30 | 23.58 | 57.36 | 63.76 | 19.74 | 54.50 | 61.20 |
| Noise rate = 60% | | | | | | | | | |
| Label correction | Bimodal | | | Symmetric | | | Flip | | |
| | 5% | 10% | 15% | 5% | 10% | 15% | 5% | 10% | 15% |
| *ce only* | **58.42** | 66.96 | **69.41** | **55.22** | **66.87** | **69.46** | **65.20** | **68.33** | **70.41** |
| *rce only* | 54.35 | **67.24** | 69.41 | 24.90 | 58.72 | 64.39 | 14.43 | 46.09 | 68.75 |
| *both* | 24.35 | 55.18 | 61.19 | 25.62 | 55.16 | 61.46 | 12.46 | 36.76 | 50.83 |

APPENDIX

## A  TRAINING WITH CORRECTED LABELS: VISION RESULTS

Here we present the extensive results of our empirical evaluation on training with corrected labels for the vision datasets. This complements the results presented in §3. We compare the impact of correcting labels only on the cross entropy term (*ce only*), only on the reverse cross entropy term (*rce only*), or *both*. Table 3 and Table 4 show the achieved accuracy for CIFAR-10 and CIFAR-100, respectively, under two noise rates, 30% and 60%, three different noise types, symmetric, bimodal and flip, and three fractions of trusted data, 5%, 10% and 15%. For each noise scenario the best case is highlighted in bold. *ce only* achieves the highest accuracy in all cases except one. Under 60% bimodal noise on CIFAR-100 with 10% trusted data *rce only* is slightly better by 0.28 percent points. More in general, *rce only* typically performs second best and *both* achieves the worst accuracy. Focusing on *ce only* over the other two, the gain tends to increase with the difficulty of the noise scenarios, i.e. with higher number of classes, higher noise rates and less trusted data. *ce only* outperforms the other two by up to 11.74 percent points for CIFAR-10 and up to 51.03 percent points for CIFAR-100.

## B  TEXT ANALYSIS ON TWITTER AND SST DATASETS WITH VARYING NOISE RATES

Figure 4 shows how the accuracy changes with respect to different noise rates on the Twitter and SST datasets. GSL and GLC are provided with one percent trusted data. In contrast, GSL can effectively use the symmetric cross entropy to overcome the limitation of low trusted data. This also explains why STMATRIX always trails closely behind GSL by using the true confusion matrix and symmetric entropy loss. One may further improve STMATRIX by using the optimal weights of $A$

and $B$ according to the true corruption matrix, instead of estimated corruption matrix of GSL. The Twitter dataset highlights well the differences (see Figure 4a). The SST dataset is an easier problem with only two classes and all methods are able to perform equally well (see Figure 4b).

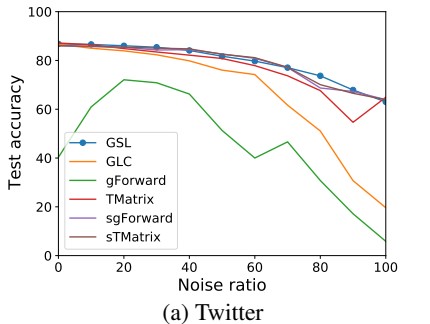
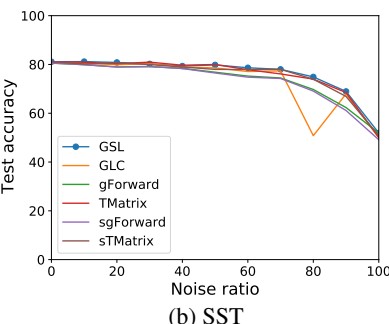

(a) Twitter $\qquad$ (b) SST

Figure 4: Testing accuracy on text datasets with varying noise rates (1% trusted data).

## C  PROOF OF ROBUSTNESS OF CROSS ENTROPY LOSS WITH LABEL CORRECTION THEOREM 4.1

**Theorem 4.1.** *In a multi-class classification problem, $\ell_{ce}$ with prediction correction is noise tolerant under symmetric label noise if the noise rate $\varepsilon < \frac{K-1}{K-\frac{\Delta\mathcal{A}}{\Delta R}}$, where $\Delta\mathcal{A} = \sum_{k=1}^{K} \ell_{ce}(\boldsymbol{C}^{*T}f(\boldsymbol{x}),k) - \sum_{k=1}^{K} \ell_{ce}(\hat{\boldsymbol{C}}^{T}f(\boldsymbol{x}),k)$, and $\Delta R$ is the difference of risk minimization between optimal classifier and $f$. And $\ell_{ce}$ with prediction correction is also noise tolerant under flip noise when noise rate $\varepsilon_{yk} \leq (1+\frac{\Delta\mathcal{W}_y}{\Delta\mathcal{W}_k}) - \varepsilon_y(1+\frac{\Delta\mathcal{W}_y}{\Delta\mathcal{W}_k})$ where $\varepsilon_k$ and $\varepsilon_{yk}$ are the correct and flipped class probabilities, respectively.*

*Proof.* For symmetric noise:

$$R_\varepsilon(f,\hat{\boldsymbol{C}}) = \mathbb{E}_{\boldsymbol{x},\tilde{y}}\ell_{ce}(\hat{\boldsymbol{C}}^T f(\boldsymbol{x}),\tilde{y}) = \mathbb{E}_{\boldsymbol{x}}\mathbb{E}_{y|\boldsymbol{x}}\mathbb{E}_{\tilde{y}|\boldsymbol{x},y}\ell_{ce}(\hat{\boldsymbol{C}}^T f(\boldsymbol{x}),y)$$

$$= \mathbb{E}_{\boldsymbol{x},y}[(1-\varepsilon)\ell_{ce}(\hat{\boldsymbol{C}}^T f(\boldsymbol{x}),y) + \frac{\varepsilon}{K-1}(\sum_{k\neq y}\ell_{ce}(\hat{\boldsymbol{C}}^T f(\boldsymbol{x}),k))]$$

$$= \mathbb{E}_{\boldsymbol{x},y}[(1-\varepsilon)\ell_{ce}(\hat{\boldsymbol{C}}^T f(\boldsymbol{x}),y) + \frac{\varepsilon}{K-1}(\sum_{k=1}^{K}\ell_{ce}(\hat{\boldsymbol{C}}^T f(\boldsymbol{x}),k) - \ell_{ce}(\hat{\boldsymbol{C}}^T f(\boldsymbol{x}),y))]$$

$$= (1-\varepsilon)R(f(\boldsymbol{x},\hat{\boldsymbol{C}}) + \frac{\varepsilon}{K-1}(\sum_{k=1}^{K}\ell_{ce}(\hat{\boldsymbol{C}}^T f(\boldsymbol{x}),k) - R(f,\hat{\boldsymbol{C}}))$$

$$= R(f,\hat{\boldsymbol{C}})(1-\frac{\varepsilon K}{K-1}) + \frac{\varepsilon}{K-1}(\sum_{k=1}^{K}\ell_{ce}(\hat{\boldsymbol{C}}^T f(\boldsymbol{x}),k))$$

$$\tag{7}$$

Let $\mathcal{A}(\hat{\boldsymbol{C}}^T f(\boldsymbol{x}),y) = \sum_{k=1}^{K}\ell_{ce}(\hat{\boldsymbol{C}}^T f(\boldsymbol{x}),k)$. Then we can rewrite (7) as $R_\varepsilon(f,\boldsymbol{C}) = (1-\frac{\varepsilon K}{K-1})R(f,\boldsymbol{C}) + \frac{\varepsilon}{K-1}\mathcal{A}(\boldsymbol{C}^T f(\boldsymbol{x}),y)$, thus:

$$R_\varepsilon(f^*,\boldsymbol{C}^*) - R_\varepsilon(f,\hat{\boldsymbol{C}}) = (1-\frac{\varepsilon K}{K-1})\underbrace{(R(f^*,\boldsymbol{C}^*) - R(f,\hat{\boldsymbol{C}}))}_{\Delta R}$$

$$+ \frac{\varepsilon}{K-1}\underbrace{(\mathcal{A}(\boldsymbol{C}^{*T}f^*(\boldsymbol{x}),y) - \mathcal{A}(\hat{\boldsymbol{C}}^T f(\boldsymbol{x}),y))}_{\Delta\mathcal{A}}$$

$$\tag{8}$$

where $\Delta R \leq 0$, because $f^*$ is the global minimizer for $R$ and $\boldsymbol{C^*}$ the optimal noise confusion matrix. Similarly, $\Delta \mathcal{A} \leq 0$ because for the optimal case we can say $\mathcal{A}(\boldsymbol{C^*}^T f^*(\boldsymbol{x}), y) \approx 0$. $\ell_{ce}$ with label correction is robust to noise when $R_\varepsilon(f^*, \boldsymbol{C^*}) - R_\varepsilon(f, \hat{\boldsymbol{C}}) \leq 0$. This is true when:

$$
\begin{aligned}
R_\varepsilon(f^*, \boldsymbol{C^*}) - R_\varepsilon(f, \hat{\boldsymbol{C}}) = (1 - \frac{\varepsilon K}{K-1})\Delta R + \frac{\varepsilon}{K-1}\Delta \mathcal{A} \\
= \Delta R - \frac{\varepsilon K}{K-1}\Delta R + \frac{\varepsilon}{K-1}\Delta \mathcal{A} \leq 0 \xRightarrow{\Delta R \leq 0} \\
1 - \frac{\varepsilon K}{K-1} + \frac{\varepsilon}{K-1}\frac{\Delta \mathcal{A}}{\Delta R} \geq 0 \Rightarrow \\
1 \geq \frac{\varepsilon}{K-1}(K - \frac{\Delta \mathcal{A}}{\Delta R}) \Rightarrow \\
\varepsilon \leq \frac{K-1}{K - \frac{\Delta \mathcal{A}}{\Delta R}}
\end{aligned}
\tag{9}
$$

With no label correction, $\boldsymbol{C}$ is missing in $\Delta \mathcal{A}$ and the two terms become equal, i.e. $\Delta \mathcal{A} = 0$. In this condition the bound becomes $\varepsilon < \frac{K-1}{K}$ as found by (Ghosh et al., 2017) for cross-entropy without label correction. Since $\frac{\Delta \mathcal{A}}{\Delta R} \geq 0$, $\varepsilon < \frac{K-1}{K} \leq \frac{K-1}{K - \frac{\Delta \mathcal{A}}{\Delta R}}$ the new bound can be seen as generalization of the previous bound. $\frac{\Delta \mathcal{A}}{\Delta R}$ should also be less than one to ensure a meaningful bound on $\varepsilon$, avoiding scenarios of noise rate greater than 1.

For asymmetric flip noise, $1 - \varepsilon_y$ is the probability of a label being correct (i.e., $k = y$), and the noise condition $\varepsilon_{yk} < 1 - \varepsilon_y$ generally states that a sample $\boldsymbol{x}$ has a higher probability $(1 - \varepsilon_y)$ of being classified correctly as class $y$, rather than the probability $(\varepsilon_{yk})$ of being classified incorrectly as class $k \neq y$.

$$
\begin{aligned}
R_\varepsilon(f, \boldsymbol{C}) = \mathbb{E}_{\boldsymbol{x}, \tilde{y}} \ell_{ce}(\boldsymbol{C}^T f(\boldsymbol{x}), \tilde{y}) = \mathbb{E}_{\boldsymbol{x}} \mathbb{E}_{y|\boldsymbol{x}} \mathbb{E}_{\tilde{y}|\boldsymbol{x}, y} \ell_{ce}(\boldsymbol{C}^T f(\boldsymbol{x}), y) \\
= \mathbb{E}_{\boldsymbol{x}, y}[(1 - \varepsilon_y)\ell_{ce}(\boldsymbol{C}^T f(\boldsymbol{x}), y) + \sum_{k \neq y} \varepsilon_{yk} \ell_{ce}(\boldsymbol{C}^T f(\boldsymbol{x}), k)] \\
= \mathbb{E}_{\boldsymbol{x}, y}[(1 - \varepsilon_y)(\sum_{k=1}^K \ell_{ce}(\boldsymbol{C}^T f(\boldsymbol{x}), k) - \sum_{k \neq y} \ell_{ce}(\boldsymbol{C}^T f(\boldsymbol{x}), y)) + \sum_{k \neq y} \varepsilon_{yk} \ell_{ce}(\boldsymbol{C}^T f(\boldsymbol{x}), k)] \\
= \mathbb{E}_{\boldsymbol{x}, y}[(1 - \varepsilon_y) \sum_{k=1}^K \ell_{ce}(\boldsymbol{C}^T f(\boldsymbol{x}), k) + \sum_{k \neq y} (1 - \varepsilon_y - \varepsilon_{yk}) \ell_{ce}(\boldsymbol{C}^T f(\boldsymbol{x}), k)]
\end{aligned}
\tag{10}
$$

Similar to the symmetric case we require that $R_\varepsilon(f^*, \boldsymbol{C^*}) - R_\varepsilon(f, \hat{\boldsymbol{C}}) \leq 0$ for the loss to be robust to noise:

$$
\begin{aligned}
R_\varepsilon(f^*, \boldsymbol{C^*}) - R_\varepsilon(f, \hat{\boldsymbol{C}}) = \mathbb{E}_{\boldsymbol{x}, y}[(1 - \varepsilon_y)(\sum_{k=1}^K \underbrace{\ell_{ce}(\boldsymbol{C^*}^T f^*(\boldsymbol{x}), k) - \ell_{ce}(\hat{\boldsymbol{C}}^T f(\boldsymbol{x}), k))}_{\Delta \mathcal{W}_y} \\
+ \sum_{k \neq y} (1 - \varepsilon_y - \varepsilon_{yk}) \underbrace{\ell_{ce}(\boldsymbol{C^*}^T f^*(\boldsymbol{x}), k) - \ell_{ce}(\hat{\boldsymbol{C}}^T f(\boldsymbol{x}), k)}_{\Delta \mathcal{W}_k}] \leq 0
\end{aligned}
\tag{11}
$$

where $\Delta \mathcal{W}_y \leq 0$ and $\Delta \mathcal{W}_k \leq 0$ because $C^*$ is the optimal noise confusion matrix. Rewriting (11):

$$\mathbb{E}_{\boldsymbol{x},y}[\sum_{k=1}^{K}(1 - \varepsilon_y)\Delta \mathcal{W}_y + \sum_{k \neq y} \varepsilon_{yk}\Delta \mathcal{W}_k] \leq 0 \Rightarrow$$

$$\sum_{k=1}^{K}(1 - \varepsilon_y)\Delta \mathcal{W}_y + \sum_{k \neq y}(1 - \varepsilon_y - \varepsilon_{yk})\Delta \mathcal{W}_k \leq 0 \Rightarrow$$

$$\Delta \mathcal{W}_y - \varepsilon_y \Delta \mathcal{W}_y \leq -\Delta \mathcal{W}_k + \varepsilon_y \Delta \mathcal{W}_k + \varepsilon_{yk}\Delta \mathcal{W}_k \xRightarrow{\Delta \mathcal{W}_k \leq 0}$$ 

$$\frac{\Delta \mathcal{W}_y}{\Delta \mathcal{W}_k} - \varepsilon_y \frac{\Delta \mathcal{W}_y}{\Delta \mathcal{W}_k} \geq -1 + \varepsilon_y + \varepsilon_{yk} \Rightarrow$$

$$\frac{\Delta \mathcal{W}_y}{\Delta \mathcal{W}_k} - \varepsilon_y(\frac{\Delta \mathcal{W}_y}{\Delta \mathcal{W}_k} + 1) \geq \varepsilon_{yk} - 1$$

(12)

According to (12), the bound is $\varepsilon_{yk} \leq (1 + \frac{\Delta \mathcal{W}_y}{\Delta \mathcal{W}_k}) - \varepsilon_y(1 + \frac{\Delta \mathcal{W}_y}{\Delta \mathcal{W}_k})$. With no label correction $\Delta \mathcal{W}_y = 0$ and the bound becomes $\varepsilon_{yk} < 1 - \varepsilon_y$ as found by prior art.

$\square$

