# OpenReview forum: "Robust Learning via Golden Symmetric Loss of (un)Trusted Labels"
_ICLR.cc/2021/Conference — Reject_

### Official Review · AnonReviewer3 · 2020-10-26
**The algorithm is simple. There are some flaws in theoretical proofs.**

**Rating:** 3
**Confidence:** 5

**Review:**

This work proposes a golden symmetric loss (GSL) for the noisy label learning problems. Compared to previous weighted symmetric cross-entropy losses, the proposed loss estimates the trade-off parameters using the transition matrix. Empirical studies demonstrate that the GSL method is better than some baselines.

Comments:
1. The proposed GLS is simple. My main problem is that why should we estimate $A(\hat{C})$ and $B(\hat{C})$ simultaneously? As they are trade-off parameters, estimating one leads to the same results, but the trainable parameters can be reduced. Moreover, it seems that GSL loss only works when the noise data are generated via a transition matrix. Would it be better if we consider the instance-dependent case and relate the trade-off parameters to the feature $x$?
2. There are some flaws and informal presentations in theoretical proofs.
- Eq (7) basically follows Proof 2 in (Ghosh et. al. 2017). But, in the first equation, the law of total expectation should be $E_{x}E_{y|x}E_{\tilde{y}|y,x}$ instead of $E_{x}E_{y|x}E_{y|\tilde{y},x}$.
- In the fourth equation, the expection should be equipped to $l_{ce}(\cdot)$, since there are $f(x)$ in $l_{ce}(\cdot)$. Hence, it is also desirable to take expection to define $\mathcal{A}(\hat{C}^Tf(x),y)$. Otherwise, $\mathcal{A}(\cdot)$ depends deeply on the instance $x$.
- The biggest problem is $\Delta\mathcal{R}$ need not be non-positive. The reason is that $f*$ is the optimal solution to $\arg\min R(\cdot,C*)$ instead of $\arg\min R(\cdot,\cdot)$. The non-positivity requires a detailed discussion, and I think this property does not hold in some conditions.
- The lower bound of $\Delta A / \Delta R$ is not discussed. If $\Delta A / \Delta R < 1$, then the algorithm is able to learn from 100% noise data, which is not realizable.
- What is the definition of noise-tolerant? The authors need to give a mathematical definition.
- It is weird that Theorem 4.1 merely analyzed the vanilla cross-entropy loss. The authors need to bridge the results of Theorem 4.1 to GLS loss.

Minor comments:
1. The references are informal. Please put the author's names into the brackets.
2. The experimental results seem to be far away from state-of-the-art noisy label learning methods, such as DivideMix [1]. While the SOTA performance is not the most essential issue for me to judge a paper, I highly recommend the authors to explore better model architectures that provide competitive results.
3. The related works are not thorough for me. Some state-of-the-art noisy label learning methods (in 2019/2020), e.g. DivideMix, are not discussed.

[1] Li J, Socher R, Hoi S C H. DivideMix: Learning with Noisy Labels as Semi-supervised Learning[C]//International Conference on Learning Representations. 2019.

---

> ### Author Response · Authors · 2020-11-17
> **Response**
>
> Thank you for your constructive comments. We address your concern below:
>
> 1.1 Tuning both $A()$ and $B()$ is redundant
>
> To get the best performance we combine using golden(trusted) data with symmetric loss. Hence the total loss includes a third term stemming from the golden data instances which use a fixed weight of 1. Hence we tune both $A()$ and $B()$.
>
> 1.2 Requires noise transition matrix
>
> The noise transition matrix is estimated from the data during the first stage of the training. Moreover, the transition matrix is a generic noise representation and can be broadly used even on real world noises, see example with the Clothing1M dataset in the paper.
>
> 2.1 Typo in Eq.7.
> Thank you for pointing it out. We corrected the typo.
>
> 2.2 Expectation in Eq.4
>
> During the first stage we train a neural network using simple cross entropy to estimate the confusion matrix $C$. Eq.4 is the per sample loss used during the second stage training where $C$ is fixed hence there is no expectation. We suppose that the confusion stems from using $f()$ in the formulas and $g()$ for the second stage in Figure 3. We updated the figure and related text to avoid this confusion.
>
> 2.3 Non-positivity of $\Delta R$
>
> We assume $f^*$ is the global minimizer for $R(.,.)$ same as (Wang Y. et al. 2019) assumption. $R(.,.)$ is a risk function which is not affected by noise, only $R_\varepsilon$, so $\Delta R < 0$ by definition. The main point is that $R$ is the risk of classifier $f$, and it has a global minimizer which is $f^*$.
>
>
> 2.4  Bounds on $\frac{\Delta \mathcal{A}}{\Delta R}$
>
> $\Delta R$ is non-positive because by definition it is the risk difference with respect to an optimal solution which minimizes the risk. $\Delta  \mathcal{A}$ is non-positive because again it is the difference with respect to an optimal solution and an optimal confusion matrix. Hence, the ratio of $\frac{\Delta  \mathcal{A}}{\Delta R}$ must be non-negative. Also $\frac{\Delta \mathcal{A}}{\Delta R}$ must be less than one to guarantee a meaningful bound on $\varepsilon$, avoiding scenarios of noise greater than 100\%. We extended the discussion in the proof.
>
> 2.5 Mathematical definition of noise tolerance.
>
> We use the noise tolerance as defined in Definition 1 in (Manwani N. et al.): [Risk minimization under loss function $L$ is said to be noise tolerant if $P[sign(f^*(x)) = y_x] = P[sign(f^*_{\varepsilon} (x)) = y_x]$, where the probability is with respect to the underlying distribution of $(x, y_x)$ ]. We added an explicit reference to this definition to the paper.
>
> References:
> (Manwani N. et al. 2013) N. Manwani and P. S. Sastry, "Noise Tolerance Under Risk Minimization," in IEEE Transactions on Cybernetics, vol. 43, no. 3, pp. 1146-1151, June 2013.
>
> 2.6 Analysis of Theorem 4.1
>
> Theorem 4.1 shows that the cross entropy loss corrected via the confusion matrix is noise tolerant under the proposed condition. The other loss terms have been proven to be noise tolerant by (Wang Y. et al. 2019) and (Ghosh et al. 2017). From the experimental results we empirically see that the combined GLC loss improves classification results.
>
> Minor:
>
> 1. We changed the format of references.
> 2. We checked the paper and added it to the related work discussion. On real world noisy data, i.e. Clothing1M, we are on par with current state-of-the-art.
> 3. We extended the related work discussion with additional papers drawn from the related work including: Junnan Li, Richard Socher, and Steven C.H. Hoi.  Dividemix: Learning with noisy labels as semi-supervised learning. In ICLR, 2020.
>
> References:
>
> (Wang Y. et al. 2019) Wang Y, Ma X, Chen Z, Luo Y, Yi J, Bailey J. Symmetric cross entropy for robust learning with noisy labels. In Proceedings of the IEEE International Conference on Computer Vision 2019 (pp. 322-330).
>
> (Ghosh et al. 2017) Aritra Ghosh, Himanshu Kumar, and P. S. Sastry. Robust loss functions under label noise for deepneural networks. In AAAI, pp. 1919–1925, 2017.

---

### Official Review · AnonReviewer1 · 2020-10-26
**Interesting direction to improve robust learning from noisy labels, but theory and experiments have a few issues**

**Rating:** 5
**Confidence:** 4

**Review:**

##### Summary

The paper proposes a golden symmetric loss method that combines cross entropy loss and reverse cross entropy loss, and at the same time performs forward loss correction for cross entropy loss, under the problem where we have noisy training labels and clean training labels. It shows theoretical results showing the robustness of cross entropy loss under forward loss correction, and shows that it can be regarded as a generalization of Ghosh et al. 2017's analysis. Empirically, it shows how loss correction on reversed cross entropy or the symmetric cross entropy performs worse compared with only cross entropy, and propose to use forward loss correction only on cross entropy term.

##### Strengths

Theoretical results show the robustness of cross entropy loss under forward loss correction and compares with Ghosh et al. 2017's analysis.

Experiments show that the proposed method is better than baselines in vision and text datasets.

##### Weaknesses

The theoretical analysis gives new insights to the properties of cross entropy loss with forward label correction, but it is only a special case of the proposed loss function in the paper when $B(\hat{C})$ is 0.

It would be nice to have the same baselines for vision and text. For example, there is STMatrix in text but not vision, there is SCL in vision but not in text.

I think I am a bit confused to why STMatrix performs worse than GSL in Table 2. Shouldn't STMatrix perform better than GSL if the only difference is whether the corruption matrix is true or estimated? Or does GSL have access to trusted data while STMatrix doesn't? Does STMatrix use the true matrix to derive the balance between the two losses?

##### Comments and suggestions

I'm guessing that a naive method would be to regard the balance between two losses in Eq.4 as an hyperparameter that can be tuned with validation data. How will that perform compared with determining the weight by Jain's fairness index and noise rate? This experiment might help to understand the additional value of the proposed weight determining procedures.

It looks like all hyper-parameters were fixed and experiments were only conducted once. It would be nice to have at least a few trials and report mean and standard deviation.

Table 2 seems to be an average over different noise rates. Did each noise rate have a similar tendency to the average for the different methods? This is reported for Twitter dataset in Fig.4 in the Appendix but would be good to see the same figure for SST.


##### AFTER RESPONSE

Thank you for answering. My concern on theory is resolved, but I still think an additional baseline of regarding it as a hyper-parameter would be helpful as a comparison, having at least a few trials with mean and standard deviation for the experiments would make the conclusions stronger, and unifying the baselines is important.

---

> ### Author Response · Authors · 2020-11-17
> **Response**
>
> Thank you for your constructive comments. We address your concerns below:
>
> Answer to weaknesses
>
> 1. Proof of corrected cross entropy only.
>
> We prove that the cross entropy with loss correction is noise tolerant. The other terms have been proven in the related work. Specially the reverse cross entropy term is proven to be noise tolerant in (Wang Y. et al. 2019).
>
> 2. Different baselines.
>
> We will uniform the baselines in the final version.
>
> 3. Why sTMatrix is not the best?
>
> sTMatrix has access to the true noise confusion matrix but uses constant weights derived from the experiments in Figure 2b instead of dynamically adapted ones. This leads to the observed lower performance of sTMatrix.
>
> 4. Weighting as hyper-parameter.
>
> We try to show that static values for weights highly depend on the noise ratio and noise pattern in Figure 2b. Using weighting function is more convenient and accurate, and can save training time over hyper-parameter tuning methods.
>
> 5. SST results from Table 2:
>
> We added the same figure for the SST as for the Twitter dataset in the appendix and updated the text.
>
> References:
>
> (Wang Y. et al. 2019) Wang Y, Ma X, Chen Z, Luo Y, Yi J, Bailey J. Symmetric cross entropy for robust learning with noisy labels. In Proceedings of the IEEE International Conference on Computer Vision 2019 (pp. 322-330).

---

### Official Review · AnonReviewer2 · 2020-10-28
**Review for "Robust Learning via Golden Symmetric Loss of (un)Trusted Labels"**

**Rating:** 4
**Confidence:** 4

**Review:**

The authors of the paper propose a novel loss function termed the golden symmetric loss to tackle the important problem of learning with noisy labels. The proposed loss function, in essence, involves the use of a corruption matrix to correct the regular cross-entropy loss and, at the same time, to estimate the relative weighting of the corrected cross-entropy loss and that of reverse cross-entropy loss.  A series of empirical experiments were conducted to demonstrate the effectiveness of the proposed method.

In general, the paper is well written and the proposed method is technically sound. Extensive experiments across different domains and applications were also conducted to demonstrate the effectiveness of the proposed method. The idea to use a confusion matrix to automatically tune for the relative weighting between cross-entropy and reverse cross-entropy is also interesting. One significant weakness of the paper is the lack of novelty. To the best of my understanding, the paper is, in most part, a mere combination of two of the previous approaches (loss correction with corruption matrix and the Symmetric Cross-Entropy). As such, it is not surprising that it works better than these previously proposed strategies. In addition, while the idea of automatically determining the relative weighting is interesting, I am not fully clear about how the function $A(\hat{C})$ and $A(\hat{C})$ are determined. It was stated that the "final weights proportional to $J$, $\epsilon$". Can you explicitly define the functions  $A(\hat{C})$ and $A(\hat{C})$? Does this mean that the exact weighting still needs to be tuned? If so, what is the point of using $A(\hat{C})$ and $A(\hat{C})$? Can't we just directly tune the weighting explicitly? I think further clarification and elaboration can be helpful. Due to the reasons mentioned above, I recommend weakly rejecting the paper.

Questions:
1. How important is the clean dataset for the algorithm? Since the corruption matrix can be estimated without a clean dataset, it would be interesting to see experiments without a clean dataset.
2. Why is the corruption matrix applied only to regular cross-entropy but not reverse cross-entropy?

---

> ### Author Response · Authors · 2020-11-17
> **Response**
>
> Thank you for your constructive comments. We address your questions below:
>
>
> To clarify the steps, we summarize the steps in the following.
>
> Intuitively, when the noise is high, say >0.5, a higher weight is given to the cross entropy than the reverse cross entropy. We explicitly consider two scenarios: the noise is greater than 0.5 and less than 0.5. In the former case, $A()$ is set to the product of Jain-index, the number of classes, and the noise ratio and $B()$ is set to the product of Jain-index, the number of classes, and non-noise ratio. The number of classes represents the difficulty of the classification problem. In case of low noise, the calculation of $A()$ and $B()$ is reversed from the case of noise greater than 0.5.
>
>
> 1. Importance of clean data set
>
> We use the clean data in two places. To estimate the confusion matrix and leverage it during training. We do not have an explicit experiment without trusted data, but our results (Table 2, Table 3 and Table 4) show the different benefits small fractions of trusted data can bring.
>
>
> 2. Why is the corruption matrix applied only to regular cross-entropy but not reverse cross-entropy?
>
> Based on the results shown in Table 3 and Table 4, this empirically holds the best results. Intuitively the confusion matrix is useful on regular cross entropy because it is weak against noisy labels however the reverse cross entropy is intrinsically more noise tolerant.

---

### Official Review · AnonReviewer4 · 2020-10-29
**an interesting work, but the novelty is not quite significant**

**Rating:** 4
**Confidence:** 3

**Review:**

###############

Summary:

This paper proposes a robust loss function that combines both cross-entropy loss and reverse cross-entropy loss.


###############

Comments:

1.	The authors are motivated by observing that GLC cannot handle hard classes effectively. However, the why GSL can handle that is not clear to me. Is that because GSL loss function contains the reverse cross-entropy loss?
2.	The contribution of this work is also not quite clear to me. As mentioned in Section 3, GSL borrows the idea from Symmetric Cross-Entropy loss from Wang et al (2019). Hence the major contribution of this work seems only proposing weights depending on the Noise Transition Matrix on the cross-entropy and reverse cross-entropy loss functions.
3.	Figure 1 is somehow hard to read. Figure 1 a) seems wrong to me. As the corruption matrix 1a) illustrates that there is no corruption. It could be better if there are more explanations for them.
4.    A() and B() -- " Final weights proportional to \epsilon and J". It is still not clear to me how to compute A and B, and why they have to be proportional to  \epsilon and J. In addition, it is not clear to me how to obtain the value of \epsilon. It is not clear to me when to estimate weights during the training process in Figure 3.
5.    The proof on page 12 seems wrong to me. To see this, the first expectation in the first line of the proof is taken over x and \tilde{y}, while the second expectation is conditional on \tilde{y}. In fact, we have p(x,\tilde{y}) = \sum_y p(\tilde{y}|y)p(y|x)p(x) (see Making Deep Neural Networks Robust to Label Noise: a Loss Correction Approach, Section 4, 2017 CVPR).
6.    Still in the proof, it is not clear to me why R_\epsilon (f^*-C^*) - R_\epsilon (f-\hat{C}) \le 0 indicates that \ell_ce with label
correction is robust to noise.


###############

Additional questions:

1.	“\alpha and \beta can significantly impact the final model accuracy”. I agree but the figure 2b) is somehow confusing to me. The values of \alpha and \beta for the orange curve are 60 times as large as those for the blue curves. And the orange curve is better. However, the corresponding optimization problems are the same, since the values of \alpha and \beta can be absorbed into the learning rate. If we use the blue curve setting and change the learning rate to 60 times as large as before, the results could be the same. The whole improvement could be resulting from changing the learning rate.
2.	The clean data only used to estimate the Noise Transition Matrix proposed by GLC method (Hendrycks et al. (2018)). Then they are added to the final loss as the clean cross-entropy loss. I was wondering if there are any other benefits of clean data that we can leverage.

###############

This paper proposes a noise-robust loss function where the contribution is leveraging the Noise Transition Matrix to estimate the weights on cross-entropy loss and reverse cross-entropy loss. It is interesting to introduce the reverse cross-entropy loss. But the novelty of this work does not meet the standard for an ICLR publication.

---

> ### Author Response · Authors · 2020-11-17
> **Response**
>
> Thank you for your constructive and valuable feedback. We address your concerns below.
>
> 1. Is GSL better due to the reverse cross-entropy loss?
>
> This is one part of the reason as can also be seen by comparing the results of gForward and sgForward. Both use the estimated noise transition matrix to correct the predictions but gForward uses the simple cross entropy (same as GLC) while sgForward adds also the reverse cross-entropy term. However, GSL outperforms even sgForward because it also adapts the two terms to the noise characteristics via dynamically weighting the two terms.
>
> 2. Contribution over Symmetric Cross-Entropy loss from (Wang Y et al. 2019).
>
> While SCL is an important piece by itself, it neither considers dynamic weighting nor correcting predictions. Both are not trivial to achieve superior performance. Weighting correctly the two terms is not easy. As shown in Figure 2b, it can lead to significant fluctuations in the classification performance. Also the best weights change for different noise patterns and ratios making dynamic weighting a crucial contribution. The weighting is also the reason why GSL is slightly better than sTMatrix which uses symmetric loss and the true, rather than estimated, noise confusion matrix. We investigated empirically different ways to apply prediction correction to find the best (Figure 2a, Table 3, and Table 4).
>
> 3. Figure 1 hard to read.
>
> We reorganized Figure 1 to underline better the difference between the two groups of subfigures and updated the text. We also corrected the color range in Figure 1a to match the range of the other plots.
>
> 4. Estimation of $\varepsilon$ and $J$ to compute $A()$ and $B()$.
>
> The first step in Figure 3 trains the neural network $g$ which is used to estimate the confusion matrix using the golden(trusted) data. From the confusion matrix we derive $\varepsilon$ and $J$. $\varepsilon$ is the noise ratio which can be computed as one minus the probability of the correct class, i.e. the diagonal element (we use the average across the classes). Cross entropy handles better low noise cases, while reverse cross entropy copes better with hard classes (Wang Y et al. 2019). $A()$ and $B()$ are used in the second step of Figure 3 to train another network $f$. We suppose some confusion derived by the fact that we previously used $f$ in the text and $g$ in Figure 3. We updated the figure and relative text to use $g$ in the first step and $f$ in the second step. We also include more details on how to compute $\varepsilon$.
>
> 5. Wrong conditional expectation on page 12.
>
> Thank you for pointing it out. We corrected the typo swapping $y$ and $\tilde{y}$.
>
> 6. Why $R_\varepsilon (f^*,C^*) - R_\varepsilon (f,\hat{C}) \le 0$ indicates that $\ell_{ce}$ with label correction is robust to noise.
>
> This stems from the definition of noise tolerance we used which states that [general learning strategy of risk minimization under a given loss function, is said to be noise-tolerant if the classifier it would learn with the noisy training data has the same probability of misclassification as that of the classifier the algorithm would learn if it is given ideal or noise-free class labels for all training data.] from (Manwani N. et al. 2013).
>
>
> Additional Questions:
>
> 1. Tuning $\alpha$ and $\beta$ at the same time seems redundant.
>
> The total loss comprises both noisy and golden(trusted) data instances. Trusted data instances use simple cross entropy with an implicit weight of one. Hence $\alpha$ and $\beta$ also influence the relative weight over the loss terms stemming from trusted data instances.
>
> 2. Possible additional uses for clean data.
>
> We use clean data to estimate the confusion matrix (and indirectly the weights for the loss terms) and leverage them during training. We agree that other ways to use it would be interesting, but did not see them yet.
>
> References:
>
> (Wang Y. et al. 2019) Wang Y, Ma X, Chen Z, Luo Y, Yi J, Bailey J. Symmetric cross entropy for robust learning with noisy labels. In Proceedings of the IEEE International Conference on Computer Vision 2019 (pp. 322-330).
>
> (Manwani N. et al. 2013) N. Manwani and P. S. Sastry, "Noise Tolerance Under Risk Minimization," in IEEE Transactions on Cybernetics, vol. 43, no. 3, pp. 1146-1151, June 2013.

---

### Public Comment · ~Ehsan_Amid1 · 2020-11-10
**Please consider referencing/comparing to these more recent works**

I would like to point out that our work (Amid et al. 2019a) extends the Generalized CE loss (Zhang and Sabuncu 2018) by introducing two temperatures t1 and t2 which recovers GCE when t1 = q and t2 = 1. Our more recent work, called the bi-tempered loss (Amid et al. 2019b) extends these methods by introducing a proper (unbiased) generalization of the CE loss and is shown to be extremely effective in reducing the effect of noisy examples. Please consider referencing/comparing to these papers.

(Amid et al. 2019a) Amid et al. "Two-temperature logistic regression based on the Tsallis divergence." In The 22nd International Conference on Artificial Intelligence and Statistics (AISTATS), 2019.

(Amid et al. 2019b) Amid et al. "Robust bi-tempered logistic loss based on Bregman divergences." In Advances in Neural Information Processing Systems (NeurIPS), 2019.

---

### Decision · Program_Chairs · 2021-01-07
**Final Decision**

**Decision:**

Reject

**Comment:**

The paper studies robust learning in the presence of noisy labels and proposes a new loss function called the golden symmetric loss (GSL) combining both regular cross-entropy and reverse cross entropy and leveraging an estimate of the corruption matrix. The paper appears to be well-written.

Pros:
- Good range of application domains (both vision and text).
- Learning with noisy labels is an important practical problem.
- Theoretical guarantees for the procedure using framework from recent work.

Cons:
- Limited novelty as the method appears to be a weighted combination of two existing ideas.
- Concerns about the baselines used: why are the same baselines not being used throughout?
- Having at least a few trials with mean and standard deviation for the experiments would make the conclusions stronger.

Overall, the limited novelty combined with the concerns about the empirical analysis was a key reason for rejection.